# BT-Chain: Bidirectional Transport Chain for Topic Hierarchies Discovery

## Abstract

Topic modeling has been an important tool for text analysis. Originally, topics discovered by a model are usually assumed to be independent. However, as a semantic representation of a concept, a topic is naturally related to others, which motivates the development of learning hierarchical topic structure. Most existing Bayesian models are designed to learn hierarchical structure, but they need non-trivial posterior inference. Although the recent transport-based topic models bypass the posterior inference, none of them considers deep topic structures. In this paper, we interpret the document as its word embeddings and propose a novel bidirectional transport chain to discover multi-level topic structures, where each layer learns a set of topic embeddings and the document hierarchical representations are defined as a series of empirical distributions according to the topic proportions and corresponding topic embeddings. To fit such hierarchies, we develop an upward-downward optimizing strategy under the recent conditional transport theory, where document information is first transported via the upward path, and then its hierarchical representations are refined according to the adjacent upper and lower layers in a layer-wise manner via the downward path. Extensive experiments on text corpora show that our approach enjoys superior modeling accuracy and interpretability. Moreover, we also conduct experiments on learning hierarchical visual topics from images, which demonstrate the adaptability and flexibility of our method.

## 1 Introduction

Topic models (TMs) like latent Dirichlet allocation (LDA) (Blei et al., 2003), Poisson factor analysis (PFA) (Zhou et al., 2012), and their various extensions (Teh et al., 2006; Hoffman et al., 2010; Blei, 2012; Zhou et al., 2016) are a family of popular techniques for discovering the hidden semantic structure from a collection of documents in an unsupervised manner. In addition to learning shallow topics, mining the potential hierarchical topic structures has obtained much research effort since the hierarchies are ubiquitous in big text corpora (Meng et al., 2020; Lee et al., 2022) and can be applied to a wide range of applications (Grimmer, 2010; Zhang et al.; Guo et al., 2020).

Hierarchical Bayesian probabilistic models have been commonly used to learn topic structures (Blei et al., 2010; Paisley et al., 2014; Gan et al., 2015; Henao et al., 2015; Zhou et al., 2016), where a hierarchy of topics are learned and the topics in the higher layers serve as the priors of the topics in the lower layers. Despite the success of Bayesian models in topic structure mining, most of them employ Bayesian posterior inference to optimize their parameters (e.g., Markov Chain Monte Carlo (MCMC) and Variational Inference (VI)), which is usually non-trivial to derive and can be less flexible and efficient for big text corpora (Zhang et al., 2018). Recent developments in Autoencoding Variational Inference (AVI) (Kingma & Welling, 2013; Rezende et al., 2014) provide stronger inference tools for Bayesian models and have inspired several neural topic models (Zhang et al., 2018; Duan et al., 2021a), resulting in improved efficiency and flexibility. However, applying AVI to neural topic models still has some limitations or concerns. First, the estimation of variational posterior always needs a trade-off between accuracy and efficiency as an asymptotically exact method (Salimans et al., 2015). Besides, the latent distributions are required to be reparameterizable and KL divergence is expected to be analytical, both of which are hard to meet for topic models since they usually depend on Dirichlet distribution or the Gamma distribution (Blei et al., 2003; Zhou et al., 2015). Another concern comes from likelihood maximization, in which the inference of topic structure relies on word

co-occurrence patterns within a document. This has been recently found to give poor quality topics in case of little evidence of co-occurrences, such as corpus with a small number of documents or containing short context (Huynh et al., 2020; Wang et al., 2022). This concern is even more acute in hierarchies learning where more topics and their correlations need to be inferred (Meng et al., 2020). Several existing studies have targeted to incorporate meta knowledge to improve topic representation. The source of side information may come from various fields, including knowledge graph (Xie et al., 2015; Duan et al., 2021b), pre-trained language model (Bianchi et al., 2021; Meng et al., 2022) and word embeddings (Wang et al., 2022; Duan et al., 2021a).

Another notable tendency developed recently is the conditional transport (CT) theory (Zheng & Zhou, 2021a). It provides an efficient tool to measure the distance between two probability distributions and has been employed in numerous machine learning problems, such as domain adaptation, generative model, and document representation (Zheng et al., 2021; Tanwisuth et al., 2021; Wang et al., 2022). The CT distance is defined by the bidirectional (forward and backward) transport cost between the source and target distributions, allowing the two distributions not to share the same support. Moreover, the CT distance can be unbiasedly approximated with the discrete empirical distributions, making it amenable to stochastic gradient descent-based optimization. Wang et al. (2022) first introduced CT into topic modeling by minimizing the transport cost between the word and topic space, resulting in better topic quality and document representation. The similar idea is shared with recent optimal transport-based methods (Kusner et al., 2015; Huynh et al., 2020; Zhao et al., 2021). However, they all focus on single-layer topic discovery, ignoring multi-level topic dependencies.

This paper goes beyond hierarchical Bayesian models for topic structure learning and aims to discover topic hierarchies based on the conditional transport between distributions. To formulate topic structure learning as a transport problem, we first provide a hierarchical, distributional view of topic modeling, where each layer of a topic hierarchy learns a set of topics presented as embedding vectors. Moreover, the to-be-learned topics share the same word embedding space and are organized in a taxonomy where the upper-level topics are more general while the lower-level topics are more specific (Zhang et al., 2018). In detail, we view each document as an empirical distribution of word embeddings and consider that a document can also be presented by the topic embeddings (Wang et al., 2022) at each layer. Those hierarchical empirical distributions have different supports but share semantic consistency across topical levels. With this view, we propose to learn topic hierarchies with a **B**idirectional **T**ransport chain (**BT-chain**) where a document's topic distributions in two adjacent layers are learned by being pushed close to each other in terms of the CT loss. This results in a more flexible and efficient method than VAI-based NTMs, while keeping the interpretability of Bayesian models.

With a different mechanism from previous hierarchical topic models, the proposed BT-chain is a straightforward and novel approach for topic structure learning, which can be flexibly integrated with deep neural networks. To achieve an efficient and end-to-end training algorithm, an upward-downward optimizing strategy is developed carefully, which first warms up the empirical distributions by transporting the input via the bottom-to-top path, and then applies the backward layer-wise refinement by considering the bidirectional information stream from the Bayesian perspective. The main contributions in this paper are as follows: (1) We view the hierarchical topic modeling from a new perspective of multi-layer conditional transport, which facilities us to develop a novel bidirectional transport chain for topic structure learning. (2) To effectively and efficiently implement the proposed method, we propose an upward-downward training algorithm for BT-chain with proper amortizations and strategy. (3) We conduct extensive experiments on text corpora to show that our approach enjoys superior modeling accuracy and interpretability compared with the state-of-the-art hierarchical topic models. To extend the application of topic modeling, we also apply it on learning hierarchical visual topics from images, which shows interesting visualizations.

## 2 BACKGROUND

In this section, we recap the background of transport distance between two discrete distributions. Let us consider two discrete probability distributions $p$ and $q \in \mathcal{P}(X)$ on space $X \in \mathbb{R}^H$: $p = \sum_{i=1}^{n} u_i \delta_{x_i}$, and $q = \sum_{j=1}^{m} v_j \delta_{y_j}$, where $x_i$ and $y_j$ are two points in the arbitrary same space $X$. $\boldsymbol{u} \in \Sigma^n$ and $\boldsymbol{v} \in \Sigma^m$, the simplex of $\mathbb{R}^n$ and $\mathbb{R}^m$, denotes two probability values of the discrete states satisfying $\sum_{i=1}^{n} u_i = 1$ and $\sum_{j=1}^{m} v_j = 1$. $\delta_x$ refers to a point mass located at coordinate $\boldsymbol{x} \in \mathbb{R}^H$.

To measure a distance between such two discrete distributions, the optimal transport (OT) distance (Villani, 2009) between $p$ and $q$ is formulated as an optimization problem: $\mathrm{OT}(p,q) = \min_{\mathbf{T}\in\mathbf{\Pi}(\boldsymbol{u},\boldsymbol{v})}\sum_{i,j}t_{ij}c_{ij}$,    $s.t.$    $\mathbf{T}\mathbb{1}^m = \boldsymbol{u}$,    $\mathbf{T}^T\mathbb{1}^n = \boldsymbol{v}$. The minimum of the transport plan $\mathbf{T}\in\mathbb{R}_{>0}^{n\times m}$ is taken over $\mathbf{\Pi}(\boldsymbol{u},\boldsymbol{v})$ with element $t_{ij}$, defined as the set of all possible joint probability measures $\pi$ on the whole space $\mathbb{R}^n\times\mathbb{R}^m$, with the marginals constraints. $\mathbb{1}^m$ is the $m$ dimensional vector of ones. $c_{ij} = c(x_i, y_j) \geq 0$ is the transport cost between the two points $x_i$ and $y_j$ defined by an arbitrary cost function $c(\cdot)$.

More recently, the demand on efficient computation and bidirectional asymmetric transport promote the development of conditional transport (CT) (Zheng & Zhou, 2021b), which can be applied to quantify the difference between discrete empirical distributions in various applications (Zheng et al., 2021; Tanwisuth et al., 2021; Wang et al., 2022). Specifically, given the above source and target distributions $p$ and $q$, the CT cost is defined with a bidirectional distribution-to-distribution transport, where a forward CT measures the transport cost from the source to the target and a backward CT reverses the transport direction. Therefore, the CT problem can be defined as:

$$\mathrm{CT}(p,q) = \min_{\overrightarrow{\mathbf{T}}\overleftarrow{\mathbf{T}}}(\sum_{i,j}\overrightarrow{t}_{ij}c_{ij} + \sum_{j,i}\overleftarrow{t}_{ji}c_{ji}),$$

where $\overrightarrow{t}_{ij}$ in $\overrightarrow{\mathbf{T}}$ acts as the transport probability (the navigator) from the source point $x_i$ to the target point $y_j$: $\overrightarrow{t}_{ij} = u_i\frac{v_je^{-d_\psi(x_i,y_j)}}{\sum_{j'=1}^m v_{j'}e^{-d_\psi(x_i,y_{j'})}}$, hence $\overrightarrow{\mathbf{T}}\mathbb{1}^m = \boldsymbol{u}$. Similarly, we have the reversed transport probability: $\overleftarrow{t}_{ji} = v_j\frac{u_ie^{-d_\psi(y_j,x_i)}}{\sum_{i'=1}^n u_{i'}e^{-d_\psi(y_j,x_{i'})}}$, and $\overleftarrow{\mathbf{T}}\mathbb{1}^n = \boldsymbol{v}$. The distance function $d_\psi(x_i, y_j)$ parameterized with $\psi$ can be implemented by deep neural networks to measure the semantic similarity between two points, making CT amenable to stochastic gradient descent-based optimization.

## 3  BT-CHAIN: BIDIRECTIONAL TRANSPORT CHAIN

We introduce the details of the proposed method, including a distributional view of multi-layer document representation, construction of BT-chain, and its upward-downward training algorithm.

### 3.1  HIERARCHICAL, DISCRETE DISTRIBUTIONAL REPRESENTATIONS OF DOCUMENTS

Given a collection of corpora with $J$ documents and $V$ distinct tokens, conventional TMs usually represent the $j$th document as a $V$ dimensional Bag-of-Word vector $\boldsymbol{x}_j \in \mathbb{R}_+^V$, where $x_{jv}$ indicates the frequency of the $v$-th word in document $j$ (Blei et al., 2003; Dieng et al., 2020). With $\boldsymbol{x}_j$ and the word embedding matrix $\mathbf{E} \in \mathbb{R}^{H\times V}$, where $H$ is the embedding dimension, we represent each document as an empirical distribution $P_j$ in the word embedding space:

$$P_j = \sum_{v=1}^V \hat{x}_{jv}\delta_{\boldsymbol{e}_v}, \quad \text{with} \quad \boldsymbol{e}_v \in \mathbb{R}^H, \tag{1}$$

where $\hat{\boldsymbol{x}}_j \in \Sigma^V$ is the normalization of $\boldsymbol{x}_j$ and $\boldsymbol{e}_v$ is the embedding of the $v$-th word in the vocabulary, i.e., the $v$-th column of $\mathbf{E}$. Notably that $P_j$ in Eq. 1 not only contains the word co-occurrence patterns but also considers word semantic information, which has been proven useful for high-quality topic learning (Dieng et al., 2020; Meng et al., 2022) but is often ignored by conventional TMs.

This paper aims to learn $L$ layers of topics in a topic hierarchy, each of which contains $K_l$ topics for satisfying $K_{l+1} < K_l < K_{l-1}$. It means that in the higher layers, there are fewer yet more abstract topics. Different from conventional TMs that assume a topic as the distribution over words, we view topics as the continuous vectors that lie in the same semantic space of words. Thus, each layer in BT-chain is associated with a set of topic embeddings $\{\boldsymbol{\alpha}_k^{(l)}\}_{k=1}^{K_l}, l = 1, ..., L$. Together with the topic proportion of $j$-th document $\boldsymbol{\theta}_j^{(l)} \in \mathbb{R}_+^{K_l}$ that denotes the topical weights over the $K_l$ topics, we derive the topical distributional representation of the $j$-th document in layer $l$ as:

$$Q_j^{(l)} = \sum_{k=1}^{K_l} \hat{\theta}_{kj}^{(l)}\delta_{\boldsymbol{\alpha}_k^{(l)}}, \quad \text{with} \quad \boldsymbol{\alpha}_k^{(l)} \in \mathbb{R}^H, \quad l = 1, ...L, \tag{2}$$

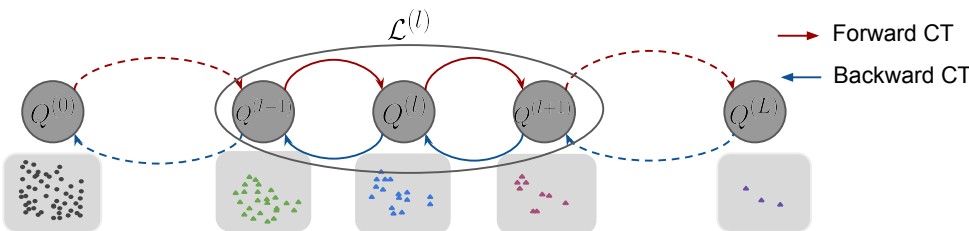

Figure 1: The directed graphical model of BT-chain. The adjacent layers are connected via the bidirectional CT, and $\mathcal{L}^{(l)}$ of BT-chain links $Q^{(l-1)}, Q^{(l)}, Q^{(l+1)}$ via the upward-downward training strategy, guaranteeing their semantic consistency. The solid dots and triangles denote word and topic embeddings in the embedding space, and different color means different layers.

where $\hat{\boldsymbol{\theta}}_j^{(l)} := \boldsymbol{\theta}_j^{(l)} / \sum_{k=1}^{K_l} \theta_{kj}^{(l)}$ is the normalized topic proportions of document $j$ at $l$-th layer that ensures the simplex constraint. We follow previous hierarchical topic models (Duan et al., 2021a) and employ an encoder network $f_\omega$ parameterized by $\omega$ to infer the multi-layer topic proportions: $\{\boldsymbol{\theta}_j^{(l)}\}_{l=1}^L = f_\omega(\boldsymbol{x}_j)$. $f_\omega$ is implemented by several stacked fully connected layers and the details can be found in Appendix. B.

To summarize, the proposed BT-chain views the $j$-th document as a series of empirical distributions over the word and topic embeddings: $\{Q_j^{(l)}\}_{l=0}^L$. With $\boldsymbol{\alpha}^{(0)} = \mathbf{E}, \boldsymbol{\theta}_j^{(0)} = \boldsymbol{x}_j, K_0 = V$, we have $Q_j^{(0)} = P_j$ which denotes the word-level observed data. Note that those empirical distributions are all defined by the word/topic weights together with the corresponding semantic vectors. They are formulated in a similar form and share semantic consistency but with different topical supports.

## 3.2 Link $\{Q_j^{(l)}\}_{l=0}^L$ Via BT-chain

Given the above distributional view of documents, we develop BT-chain to learn hierarchical document distributions $\{Q_j^{(l)}\}_{l=1}^L$, where each layer in BT-chain is linked with its adjacent layers via the conditional transport and corresponds to learning of topic embeddings and topic proportions. Fig. 1 shows the overview of BT-chain. Specifically, in the topic hierarchy $l = 1, ... L - 1$, the current layer $l$ is connected with its lower layer $l - 1$ and higher layer $l + 1$ (the uppermost layer $L$ is only connected to the lower layer $L - 1$). Note that $Q_j^{(l-1)}, Q_j^{(l)}, Q_j^{(l+1)}$ capture the semantics of the same document, where $Q_j^{(l-1)}$ focuses on more detailed semantics while $Q_j^{(l+1)}$ attends to more general concepts. $Q_j^{(l)}$ thus acts here as an intermediate node that integrates information from both directions and learns semantically smooth topics. To meet such properties, it is natural to push $Q_j^{(l)}$ to $Q_j^{(l-1)}$ and $Q_j^{(l+1)}$ as close as possible. It poses a question on how to define the *closeness* between two discrete distributions with different supports. Although recent studies have provided several transport-based alternatives to measure such distance (Cuturi, 2013; Yurochkin et al., 2019; Hu et al., 2021; Zhao et al., 2021), most of them focus on shallow transport problems and ignore the deep case. To this end, our proposed BT-chain derives a layer-wise distance with the recent CT technique and expresses the transport loss in $l$-th layer as:

$$\mathcal{L}^{(l)}\left(Q_j^{(l-1)}, Q_j^{(l)}, Q_j^{(l+1)}\right) = \text{CT}\left(Q_j^{(l-1)}, Q_j^{(l)}\right) + \text{CT}\left(Q_j^{(l)}, Q_j^{(l+1)}\right). \tag{3}$$

$\mathcal{L}^{(l)}$ links the adjacent three layers via the two CT costs to guarantee the current layer $Q_j^{(l)}$ semantically close to $Q_j^{(l-1)}$ and $Q_j^{(l+1)}$. Thus for a hierarchy with $L$ layers, we have $\{\mathcal{L}^{(l)}\}_{l=1}^{L-1}$, each of which strengthens the learning of its local $Q_j^{(l)}$ by considering the neighborhood information and they together ensure the semantic consistency of the transport chain. Moreover, we minimize the above loss in terms of $Q_j^{(l)}$, which by definition consists of $\hat{\boldsymbol{\theta}}_j^{(l)}$ and $\{\boldsymbol{\alpha}_k^{(l)}\}_{k=1}^{K_l}$. The detailed parameterizations and implementations are shown in Section A of the Appendix and our designed training strategy for the transport chain is described in Section 3.3.

---

**Algorithm 1** Training algorithm for our proposed BT-chain.

---

**Input**: documents, pre-trained word embeddings $\mathbf{E}$, topic list $\{K_l\}_{l=1}^L$, hyperparameter $\beta$.
**Initialize**: topic embeddings $\{\boldsymbol{\alpha}_k^{(l)}\}_{k=1,l=1}^{K_l,L}$, $\omega$ of the inference network.
**for** iter = 1,2,3,... **do**
    Sample a batch of $B$ documents and get $\{Q_j^{(0)}\}_{j=1}^B$ with Eq. (1)
    Get $\{\boldsymbol{\theta}_j^{(l)}\}_{j=1,l=1}^{B,L}$ by the encoder network $f_\omega$
    # Upward warming-up
    **for** $l = 1, 2, \cdots, L$ **do**
        Fix $Q_{1:B}^{(l-1)}$ and compute $\mathcal{L}^{\text{up}} = \sum_{j=1}^B \text{CT}^{\text{reg}}\left(Q_j^{(l-1)}, Q_j^{(l)}\right)$
        Update $\{\boldsymbol{\alpha}_k^{(l)}\}_{k=1}^{K_l}$ and $\omega$ with stochastic gradients of $\mathcal{L}^{\text{up}}$
    **end for**
    # Downward refining
    **for** $l = L-1, \cdots, 1$ **do**
        Fix $Q_{1:B}^{(l+1)}$ and $Q_{1:B}^{(l-1)}$
        Compute $\mathcal{L}^{\text{down}} = \sum_{j=1}^B \text{CT}^{\text{reg}}\left(Q_j^{(l-1)}, Q_j^{(l)}\right) + \text{CT}^{\text{reg}}\left(Q_j^{(l)}, Q_j^{(l+1)}\right)$
        Update $\{\boldsymbol{\alpha}_k^{(l)}\}_{k=1}^{K_l}$ and $\omega$ with stochastic gradients of $\mathcal{L}^{\text{down}}$
    **end for**
**end for**

---

As suggested in the previous works (Zhao et al., 2021), we add a regularization term based on the cross entropy for the CT loss in Eq. 3:

$$\text{CT}^{\text{reg}}\left(Q_j^{(l)}, Q_j^{(l+1)}\right) = \text{CT}\left(Q_j^{(l)}, Q_j^{(l+1)}\right) - \beta \hat{\boldsymbol{\theta}}_j^{(l)} \log\left(\boldsymbol{\Phi}^{(l+1)} \boldsymbol{\theta}_j^{(l+1)}\right), \tag{4}$$

where $\boldsymbol{\Phi}_k^{(l+1)} = \text{Softmax}(\boldsymbol{\Psi}^{(l)} \boldsymbol{\alpha}_k^{(l+1)})$; $\boldsymbol{\Psi}^{(l)} \in \mathbb{R}^{K_l \times H}$ each row of which is $\boldsymbol{\alpha}_k^{(l)}$; $\beta$ is the trade-off hyperparameter that balances the weight of the cross entropy regularization. $\boldsymbol{\Phi}^{(l+1)}$ acts as the "decoder" that decodes $\boldsymbol{\theta}_j^{(l+1)}$ to layer $l$ and the cross entropy measures how close the decoded one to its "target" $\boldsymbol{\theta}_j^{(l)}$. In the training, we replace the CT distances in Eq. (3) with the regularized ones.

## 3.3 Upward-downward Training Algorithm for BT-chain

Given the training documents and pre-trained word embeddings $\mathbf{E}$, we aim to learn topic embeddings $\{\boldsymbol{\alpha}_k^{(l)}\}_{k=1,l=1}^{K_l,L}$ and the encoder network that infers the topic proportions $\{\boldsymbol{\theta}^{(l)}\}_{l=1}^L$. At layer $l$, the loss in Eq. (3) consists of two CT distances that connect layer $l$ with the lower and higher layers. Simultaneously minimizing such two CT distances can be difficult, so we propose a layer-wise upward-downward training algorithm, which consists of two steps in each training iteration.

**Upward Warming-up** In this step, we start with the learning of $Q_j^{(1)}$ given the data $Q_j^{(0)}$ in word embedding space, by minimizing $\text{CT}^{\text{reg}}\left(Q_j^{(0)}, Q_j^{(1)}\right)$, i.e., the first CT distance in Eq. (3). After $Q_j^{(1)}$ is learned, we fix it and then learn $Q_j^{(2)}$ by minimizing $\text{CT}^{\text{reg}}\left(Q_j^{(1)}, Q_j^{(2)}\right)$. In this way, we transport the raw information from the observed word layer to the uppermost topic layer $Q_j^{(L)}$ step by step.

**Downward refining** In the upward training step, the lower layer $l-1$ is fixed and treated as the data for its higher layer $l$, i.e., the message flows in the bottom-up direction. This does not consider how the higher layer $l$ affects its lower layer $l-1$. Therefore, after the upward step, we further introduce the downward step, where we start with the second uppermost layer $L-1$. To learn $Q_j^{(L-1)}$, we fix its two adjacent layers $Q_j^{(L)}$ and $Q_j^{(L-2)}$ and minimize $\mathcal{L}^{(l)}$, i.e., both terms in Eq. (3). Similarly, we can update the distributions until we reach layer 1. In this way, the message of both layers $l+1$ and $l-1$ will flow to layer $l$. The training algorithm is outlined in Algorithm 1.

## 3.4 Discussions

**Topic hierarchies learned by BT-chain** BT-chain aims to learn a topic hierarchy where the topics in the higher layers are more general and abstract than those in the lower layers. How BT-chain achieves this can be interpreted from an information compression or abstraction perspective. Specifically, in

layer 0, the data $Q_j^{(0)}$ lies in a $K_0$-dimensional word embedding space, which is a sparse representation of the semantics of document $j$. $Q_j^{(1)}$ lies in a $K_1$-dimensional topic embedding space, where $K_1 < K_0$. By pushing $Q_j^{(1)}$ to $Q_j^{(0)}$ as close as possible in terms of the CT cost, BT-chain forces $Q_j^{(1)}$ to compress or abstract the information in $Q_j^{(0)}$ with a denser representation. Similar things happen in the higher layers, as $K_1 > K_2 > \cdots > K_L$. Moreover, as all the topics are presented as embeddings, the topic correlations between the topics of two layers can be simply obtained by the distances between the topic embeddings, i.e., $\mathbf{\Phi}^{(l)}$ in Eq. (4). Therefore, the topic correlations in BT-chain can be interpreted in the same way as conventional hierarchical TMs (Blei et al., 2010; Zhou et al., 2016).

**Bayesian Flavor of BT-chain** Many deep topic models are implemented based on hierarchical Bayesian probabilistic models (Blei et al., 2010; Paisley et al., 2014; Gan et al., 2015; Henao et al., 2015; Zhou et al., 2016; Zhao et al., 2018), where the topics in the higher layers serve as the priors of the topics in the lower layers. In these Bayesian models, according to Bayes' theorem, a topic's posterior distribution consists of two terms: the data distribution (i.e., the word information in a document) and prior distribution (i.e., the topics in the higher layer). Although BT-chain is not a Bayesian model, it is interesting to interpret our method with a Bayesian flavor. For example, the value of $Q_j^{(1)}$ is learned according to $Q_j^{(0)}$ and $Q_j^{(2)}$, where $Q_j^{(0)}$ and $Q_j^{(2)}$ can be viewed as data and prior, respectively. Instead of learning the model with Bayes' theorem, we optimize BT-chain by minimizing the CT costs between the distributions, avoiding non-trivial Bayesian posterior inference.

## 4 RELATED WORK

**Topic structure learning** There is a surge of research interest in capturing the correlations among topics and generating topic structures. For example, there are many models based on hierarchical Bayesian prior such as the Dirichlet process (DP) and Chinese Restaurant Process (CRP), including hLDA (Griffiths et al., 2003), nCRP (Blei et al., 2010) and nHDP (Paisley et al., 2014). Li & McCallum (2006) propose the Pachinko Allocation Model (PAM) to model the co-occurrences of topics via a directed acyclic graph. hPAM (Mimno et al., 2007) is built on PAM and represents the topic hierarchical structure through the Dirichlet-multinomial parameters of the internal node distributions. More recently, various hierarchical extensions of Poisson factor analysis (PFA) (Zhou et al., 2012) have been proposed, including DPFA (Gan et al., 2015), DPFM (Henao et al., 2015), GBN (Zhou et al., 2016), and DirBN (Zhao et al., 2018). Zhang et al. (2018) develop Weibull hybrid autoencoding inference (WHAI) for GBN and Duan et al. (2021a) introduce word embedding into GBN and design the SawETM Connection (SC) to explore the relationship between topics. Although both WHAI and SawETM are the most related works to ours, which are Bayesian generative models and learned by maximizing the evidence lower bound (ELBO), our proposed model views deep topic modeling as a cross-layer transport problem.

**Topic models based on transport** Using transport between distributions is a recent trend in topic modeling. Xu et al. (2018) propose distilled Wasserstein learning (DWL) where the distance between topics is achieved by the OT between their word distributions built on the embedding-based underlying distance. The OT based LDA (OTLDA) (Huynh et al., 2020) shares a similar idea but aims to minimize the OT distance between documents and topics in the vocabulary space. Yurochkin et al. (2019) view the documents as the distributions over topics, those topics themselves are modeled as the distribution over words and introduce the hierarchical OT as the meta-distance between documents. However, it is not a topic model but a method using topic models to compute document distances. Based on topic embeddings, Zhao et al. (2021) propose the Neural Sinkhorn Topic Model (NSTM) that optimizes the OT distance between the normalized BoW vector and the topic proportions, where the cost matrix is calculated by the word and topic embeddings. In Wang et al. (2022), WeTe views each document as a set of mixtures of word embeddings and a set of mixtures of topic embeddings and employs the conditional transport (CT) cost to quantify the difference between those two sets. Compared with the above shallow topic models, our proposed model aims to learn hierarchical document representations and topic structures.

## 5 EXPERIMENTS

We in this section conduct comprehensive experiments on various datasets and compare the proposed BT-chain with different baseline methods to illustrate its superior performance.

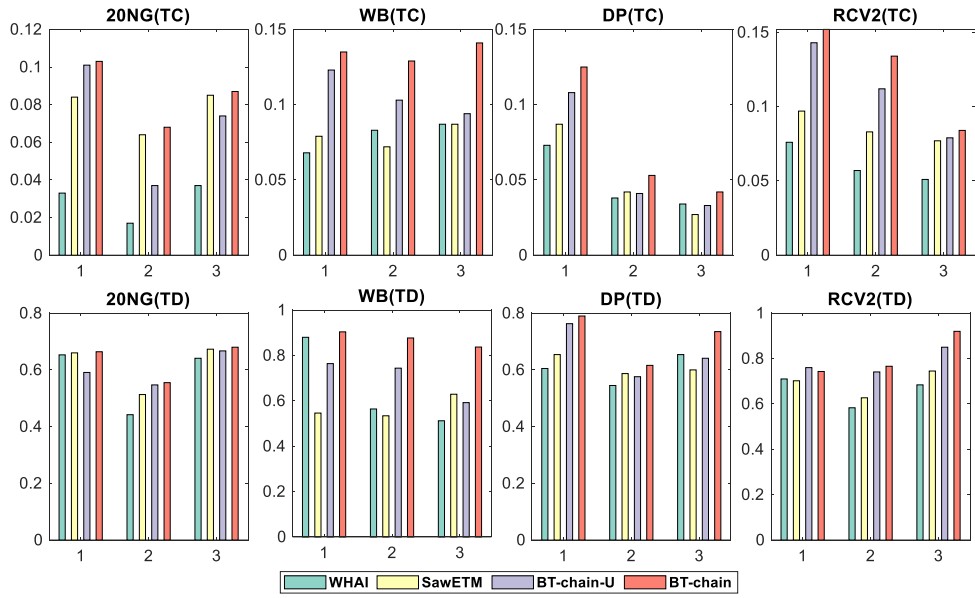

Figure 2: Topic Coherence (TC) and Topic Diversity (TD) of hierarchical topic models on the four datasets.

Table 1: Comparison of K-Means clustering purity (km-Purity) and NMI (km-NMI) for various methods. The number in brackets after the method indicates the total number of topic layers $L$. The best and second best scores of each dataset are highlighted in boldface and with an underline, respectively.

| Method | km-Purity(%) | | | | km-NMI(%) | | | |
|---|---|---|---|---|---|---|---|---|
| | WS | RCV2 | DP | 20NG(6) | WS | RCV2 | DP | 20NG(6) |
| LDA-Gibbs | 46.4±0.6 | 52.4±0.4 | 60.8 ±0.5 | 59.2±0.6 | 25.1±0.4 | 38.2±0.5 | 54.7 ±0.3 | 32.4 ±0.4 |
| DVAE | 26.6±1.5 | 52.6±1.2 | 67.2 ±1.1 | 64.6 ±1.0 | 3.7 ±0.8 | 31.3±0.9 | 50.8 ±0.6 | 29.8 ±0.6 |
| ETM | 32.9±2.3 | 50.2±0.6 | 63.1 ±1.5 | 62.6 ±2.2 | 12.3±2.3 | 30.3±1.0 | 53.2 ±0.7 | 29.3 ±1.5 |
| NSTM | 42.1±0.6 | 53.8±1.0 | 20.2 ±0.7 | 62.6±1.2 | 17.4 ±0.6 | 36.8±0.3 | 6.63±0.11 | 31.1 ±1.2 |
| WeTe | 60.8±0.2 | 62.9±0.5 | 77.1 ±1.0 | 68.5 ±0.2 | 34.9±0.4 | 42.8±0.3 | 63.7±0.4 | 36.3 ±0.2 |
| WHAI(1) | 50.8 ±0.2 | 59.0 ±0.1 | 65.1 ±0.1 | 60.4 ±0.3 | 25.8 ±0.2 | 40.0 ±0.1 | 53.0 ±0.3 | 29.6 ±0.1 |
| SawETM(1) | 34.4 ±0.4 | 62.7 ±0.4 | 65.9 ±0.2 | 70.1 ±0.2 | 10.7 ±0.6 | 44.4 ±0.3 | 54.0 ±0.3 | 38.4 ±0.2 |
| BT-chain(1)(Ours) | 61.2 ±0.2 | 62.4 ±0.3 | 77.5 ±0.3 | 70.4 ±0.1 | 35.4 ±0.4 | 44.3 ±0.2 | 66.2 ±0.2 | 39.7 ±0.1 |
| WHAI(3) | 52.1 ±0.4 | 60.5 ±0.2 | 66.9 ±0.1 | 60.8 ±0.1 | 29.9 ±0.5 | 40.0 ±0.2 | 55.2 ±0.3 | 30.2 ±0.1 |
| SawETM(3) | 43.7 ±0.8 | **64.2 ±0.5** | 70.1 ±0.1 | 72.1 ±0.3 | 21.9 ±0.7 | **45.2 ±0.4** | 58.9 ±0.2 | 42.7 ±0.1 |
| BT-chain(3)(Ours) | **62.7±0.3** | 63.7 ±0.3 | **78.0 ±0.2** | **73.3 ±0.1** | **35.9 ±0.4** | 44.6 ±0.3 | **66.6 ±0.1** | **43.9 ±0.1** |

**Datasets** We conduct the experiments on four widely used benchmark corpus: 20 News Group (20NG), Web Snippets (WS) (Phan et al., 2008), DBpedia (DP) (Lehmann et al., 2015) and Reuters Corpus Volume 2 (RCV2) (Lewis et al., 2004). These datasets have very different characteristics in terms of document length, the number of documents, and vocabulary size, where DP and RCV2 are large-scale datasets, WS and DP consist of short documents. We pre-process those datasets following WeTe (Wang et al., 2022), e.g., we tokenize and clean text by excluding standard stop words and low-frequency words. The statistics of the datasets are summarized at Table. C. 1 of Appendix.

**Baselines and settings** We compare BT-chain against conventional and advanced topic models, including *(1) Single-layer baselines*: Collapsed Gibbs Sampling LDA (**LDA-Gibbs**) as described in (Griffiths & Steyvers, 2004); Neural topic models, such as Dirichlet VAE (**DVAE**) (Burkhardt & Kramer, 2019) and embedded topic model (**ETM**) (Dieng et al., 2020), the former is a VAE based topic model and ETM is the first NTM that introduces word embeddings; NTMs with transport, Neural Sinkhorn Topic model (**NSTM**) (Zhao et al., 2021) and (**WeTe**) (Wang et al., 2022), both of them use transport distance as the loss function; *(2) Hierarchical topic models*: **WHAI** (Zhang et al., 2018) and **SawETM** (Duan et al., 2021a), they are hierarchical generative model based on Gamma belief network and are compared as our multi-layer baselines. Besides the above baselines, we also propose a variant of BT-chain that only uses the upward warming-up path to train the topic embeddings and the encoder, which we name BT-chain-U. For all baselines, we use the default parameters given with the source code or the best settings reported in their paper. For the models

Table 2: Topic coherence (TC) and topic diversity (TD) results of single-layer methods on four datasets.

| Corpus | metrics | LDA-Gibbs | DVAE | ETM | NSTM | WeTe | BT-chain |
|--------|---------|-----------|------|-----|------|------|----------|
| 20NG(6) | TC | 0.058 | 0.013 | 0.019 | 0.103 | 0.101 | **0.103** |
|         | TD | 0.631 | 0.661 | 0.551 | 0.647 | 0.591 | **0.664** |
| RCV2   | TC | 0.097 | -0.032 | 0.067 | 0.118 | 0.143 | **0.152** |
|         | TD | 0.641 | 0.730 | 0.502 | 0.639 | **0.760** | 0.743 |
| DP     | TC | 0.074 | 0.065 | 0.053 | 0.104 | 0.108 | **0.125** |
|         | TD | 0.684 | 0.645 | 0.583 | 0.715 | 0.763 | **0.790** |
| WS     | TC | 0.071 | 0.078 | 0.003 | 0.122 | 0.123 | **0.135** |
|         | TD | 0.760 | 0.540 | 0.585 | **0.940** | 0.764 | 0.904 |

that work with word embeddings, including ETM, NSTM, WeTe, and our BT-chain, we use the pre-trained GloVe vectors for a fair comparison. We summarize those baselines at Table. C. 2.

**Evaluation metrics** Notably, our work aims to mine high-quality topic structures, where evaluation metrics about topics are our main focus. Though perplexity is a common-used metric for generative topic model, recent studies suggest that it might not be an appropriate measure of the topic quality (Chang et al., 2009). Besides, CT based topic models (*e.g.*, NSTM and WeTe) are learned by minimizing the transport cost instead of maximizing the log-likelihood, which is important to achieve better perplexity results. Therefore, we put more attention on following metrics. We use Topic Coherence (TC) and Topic Diversity (TD) to evaluate the learned topics from both the interpretability and diversity aspects. In detail, TC is the average Normalized Pointwise Mutual Information (NPMI) over the top 10 words of each topic which is highly correlated to human judgment. TD is calculated by the average percentage of unique words in the top 25 words of all topics. The higher the better for both TC and TD. Besides the topic quality, we also report Purity and Normalized Mutual Information (NMI) (Schütze et al., 2008) on clustering tasks to measure the performance of document representation. We first train the model on the training datasets and infer the topic proportions on the testing documents. Given the collection of topic proportions, we apply KMeans to predict the document label. The clustering number is set as 20 for 20NG, WS, and DP, while 52 for the RCV2 dataset. For 20NG dataset, we follow WeTe and use its six super categories at the first level as the ground truth, and denote it as 20NG(6).

**Implementation details of BT-chain** As discussed above, the proposed model takes the bag of features $Q_j^{(0)} = \sum_{k=1}^{K_0} \hat{\boldsymbol{\theta}}_{kj}^{(0)} \delta_{\boldsymbol{\alpha}_k^{(0)}}$ as its input. For corpus, $\boldsymbol{\alpha}_k^{(0)}$ is the word embeddings in the vocabulary, $\hat{\boldsymbol{\theta}}_j^{(0)}$ is the normalized TF-IDF vector. We set $L$ as 3 and the topic numbers at each layer for HTMs in the baselines and BT-chain $K = [100, 64, 32]$ for a fair comparison. We set $K = 100$ for all the single-layer models. We use the Adam optimizer with learning rate 0.001, batch size 500. All experiments are performed on the same machine and our model is implemented in PyTorch.

**Quantitative results** For all the methods, we run the algorithm five times with different random seeds, and report the mean and standard deviation on the clustering task, while choosing the best results for topic quality. We fix the hyperparameter $\beta = 50$ for all the datasets and also report the effect of various settings on the document clustering in the Appendix. C.4. Table 1 reports the km-Purity and km-NMI on four corpora. The first group is the single-layer topic models, including LDA, NTMs, and transport based models, and the last two groups are hierarchical topic models (HTMs) with a single layer and three layers respectively. We find that *i)* In general, our proposed BT-chain outperforms the others in most cases, especially on the short WS and DP corpus. We attribute this to the empirical distribution representation with word/topic embeddings. *ii)* The deep models achieve better performance on both Purity and NMI than the shallow models. This is mainly because their hierarchical document representations can provide more comprehensive information which is one of our motivations. *iii)* Our single-layer BT-chain achieves better scores than WeTe, which shows the efficiency of introducing the word weights at layer 0, e.g., the normalized TF-IDF vector, rather than treating them according to the word frequency in WeTe.

To fully compare the qualities of learned topics, Table 2 and Figure 2 (where x-axis denotes the topic layers.) report the topic coherence (TC) and topic diversity (TD) for single-layer models and deep models respectively. Overall, we observe that the topics learned from our BT-chain has a better TC and TD, especially for the short corpora. This meets with the observation on the clustering task and

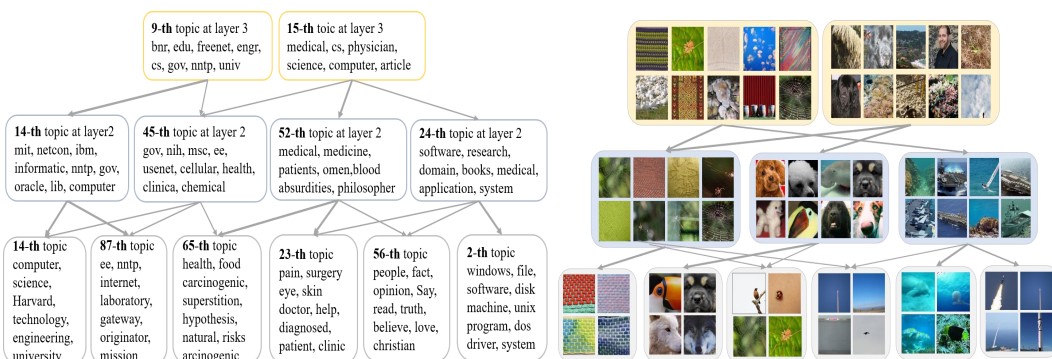

Figure 3: Topic hierarchies learned from 20NG (left) and miniImageNet (right). The links between two adjacent layers are obtained according to the semantic similarities of topics (*e.g.* $\Phi$ in Eq. 4)

again proves the efficiency of the introduced word/topic semantics. Besides, our three-layer BT-chain outperforms all the other HTMs in most cases, which benefits from the new view of topic hierarchies discovery from the transport theory. The proposed BT-chain guarantees semantic consistency by minimizing the total transport cost via $\mathcal{L}^{(l)}, l = 1, ..., L-1$, resulting in high-quality topic structures. Finally, our BT-chain consistently achieves higher scores than BT-chain-U on all datasets at high layers. Note that BT-chain-U only uses the upward path to transport messages from word space to higher-level topic space. This demonstrates the efficiency of our downward training strategy.

**Qualitative results** To visualize the learned topic structures, we show the topic hierarchies on 20NG in Fig. 3 (left), where it can be observed that topics in higher layers are mixtures of semantically close topics in the lower levels. Recalling that BT-chain receives a set of feature embeddings as its input, this broadens the scope of application of BT-chain beyond text corpora. Here we conduct experiments on miniImageNet and visualize the learned concept hierarchy in Fig. 3 (right). MiniImageNet contains a total number of 100 classes with 600 images in each class, which are extracted from the ImageNet dataset (Russakovsky et al., 2015). To obtain the empirical distribution for image data, we adopt ConceptTransformer (Rigotti et al., 2022) to obtain the bag of features for images, where the $j$-th image is first divided evenly into $N$ patches $x_j = \sum_{n=1}^{N} \frac{1}{N}\delta_{\boldsymbol{e}_{nj}}$, and each patch aligns to $M$ concepts via the cross-attention $\boldsymbol{e}_n = \sum_{m=1}^{M} \alpha_{nm}\delta_{\boldsymbol{c}_m}$, where $\boldsymbol{e}_n \in R^d$ is the patch embeddings, and $\alpha_{nm}$ is the attention weights, $\mathbf{C} \in R^{d \times M}$ is the concept embedding matrix. We average all patches and thus the final $Q_j^{(0)}$ for image can be expressed as: $Q_j^{(0)} = \sum_{m=1}^{M}(\frac{1}{N}\sum_{n=1}^{N} \alpha_{nm})\delta_{\boldsymbol{c}_m}$. Once trained on image data, we visualize the images assigned to the learned visual topics (Fig. 3 right), where we can observe interesting semantic relations between visual concepts. For example, the bottom layer contains the clear and concrete concepts, e.g., the regular textures, animal eyes, small objects in pure background, and so on. The second layer contains more complex semantics: the facial part of animal, and the small objects in textured background. It is interesting to see that the topic of warships in the second layer is closely related to the topic of oceans and the topic of missiles in the bottom layer. Similar observation can be found in the topic hierarchy of documents (Fig. 3 left), where we list the most related words of each topic. We report more qualitative results and time complexity analysis in Appendix C.3 and D.

## 6 CONCLUSION

In this paper, we present a bidirectional transport chain (BT-chain) for hierarchical representation. BT-chain views each document as a series of empirical distributions over word/topic embeddings, each of which consists of topic proportions and the corresponding embedding vectors. With the fact that those multi-layer representations share semantic consistency of the same document, we propose an effective upward-downward layer-wise training algorithm to learn topic hierarchies and document representations based on the conditional transport theory. Our framework can be viewed as a hierarchical extension to the recently developed topic models based on transport. Extensive experiments on text corpora show that our approach enjoys superior modeling accuracy and interpretability. Moreover, we also have conducted experiments on learning hierarchical visual topics from images, which demonstrate the adaptability and flexibility of our method.

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

# A   DETAILED DERIVATION OF CT IN BT-CHAIN

$CT(Q_j^{(l)}, Q_j^{(l+1)})$ measures the transport cost of two discrete empirical distributions (Zheng & Zhou, 2021b), which consists of a forward CT that constructs a navigator to transport $P_j^{(l)}$ to $P_j^{(l+1)}$ and a backward CT that reverse the transport direction:

$$\mathrm{CT}\left(Q_j^{(l)}, Q_j^{(l+1)}\right) = \overrightarrow{\mathrm{CT}}\left(Q_j^{(l)}, Q_j^{(l+1)}\right) + \overleftarrow{\mathrm{CT}}\left(Q_j^{(l+1)}, Q_j^{(l)}\right).$$

Specifically, the forward CT tries to minimize the expected transport cost from a set of $K_l$ embeddings to a set of $K_{l+1}$ embeddings:

$$\overrightarrow{\mathrm{CT}} = \mathbb{E}_{\boldsymbol{\alpha}_k^{(l)} \sim Q_j^{(l)}} \mathbb{E}_{\boldsymbol{\alpha}_k^{(l+1)} \sim \pi\left(\boldsymbol{\alpha}_k^{(l+1)} | \boldsymbol{\alpha}_k^{(l)}\right)} \left[c\left(\boldsymbol{\alpha}_k^{(l)}, \boldsymbol{\alpha}_k^{(l+1)}\right)\right], \pi\left(\boldsymbol{\alpha}_k^{(l+1)} | \boldsymbol{\alpha}_k^{(l)}\right) = \frac{\hat{\theta}_{kj}^{(l+1)} s\left(\boldsymbol{\alpha}_k^{(l)}, \boldsymbol{\alpha}_k^{(l+1)}\right)}{\sum_{k'=1}^{K_{l+1}} \hat{\theta}_{k'j}^{(l+1)} s\left(\boldsymbol{\alpha}_k^{(l)}, \boldsymbol{\alpha}_{k'}^{(l+1)}\right)},$$

where we specify transport cost $c(\cdot)$ of two embedding vectors with the inner product: $c(\boldsymbol{\alpha}_k^{(l)}, \boldsymbol{\alpha}_k^{(l+1)}) = \exp(-\boldsymbol{\alpha}_k^{(l)T} \boldsymbol{\alpha}_k^{(l+1)})$, e.g., the closer the two vectors are, the smaller the point-to-point transport cost. The navigator $\pi(\cdot)$ is the conditional probability of a given embedding $\boldsymbol{\alpha}_k^{(l)}$ being transported to embedding $\boldsymbol{\alpha}_k^{(l+1)}$, which is determined by both the topic weights $\boldsymbol{\theta}_{kj}^{(l+1)}$ and the similarity score $s(\cdot)$ of the source and target topics. Therefore, it would be easier to transport $\boldsymbol{\alpha}_k^{(l)}$ to $\boldsymbol{\alpha}_k^{(l+1)}$, if $\boldsymbol{\alpha}_k^{(l+1)}$ describes a more popular topic that is semantically closer to $\boldsymbol{\alpha}_{k_l}^{(l)}$ in the embedding space. We here also use the inner product to define $s(\boldsymbol{\alpha}_k^{(l)}, \boldsymbol{\alpha}_k^{(l+1)}) = \exp(\boldsymbol{\alpha}_k^{(l)T} \boldsymbol{\alpha}_k^{(l+1)})$ to reduce the computational cost, although other possible choices.

Similarly, we can derive the backward CT by exchanging the source set and the target set:

$$\overleftarrow{\mathrm{CT}} = \mathbb{E}_{\boldsymbol{\alpha}_k^{(l+1)} \sim Q_j^{(l+1)}} \mathbb{E}_{\boldsymbol{\alpha}_k^{(l)} \sim \pi\left(\boldsymbol{\alpha}_k^{(l)} | \boldsymbol{\alpha}_k^{(l+1)}\right)} \left[c\left(\boldsymbol{\alpha}_k^{(l+1)}, \boldsymbol{\alpha}_k^{(l)}\right)\right], \pi\left(\boldsymbol{\alpha}_k^{(l)} | \boldsymbol{\alpha}_k^{(l+1)}\right) = \frac{\hat{\theta}_{kj}^{(l)} s\left(\boldsymbol{\alpha}_k^{(l+1)}, \boldsymbol{\alpha}_k^{(l)}\right)}{\sum_{k'=1}^{K_l} \hat{\theta}_{k'j}^{(l)} s\left(\boldsymbol{\alpha}_k^{(l+1)}, \boldsymbol{\alpha}_{k'}^{(l)}\right)}.$$

# B   INFERENCE NETWORK FOR TOPIC PROPORTIONS

To guarantee the sparse property of the topic proportion $\boldsymbol{\theta}_j^{(l)}$ and as suggested in Wang et al. (2022), we leverage a Hierarchical Weibull Reparameterization Encoder (HWRE) to infer $\boldsymbol{\theta}_j^{(l)}$ from the information of the previous layers:

$$\boldsymbol{\theta}_j^{(l)} \sim \text{Weibull}\left(\boldsymbol{k}_j^{(l)}, \boldsymbol{\lambda}_j^{(l)}\right), \boldsymbol{k}_j^{(l)}, \boldsymbol{\lambda}_j^{(l)} = \text{SoftPlus}\left(g(\boldsymbol{h}_j^{(l)})\right), \quad \boldsymbol{h}_j^{(l)} = f\left(\boldsymbol{h}_j^{(l-1)}\right), \quad (5)$$

where $\boldsymbol{h}_j^{(0)} = \boldsymbol{x}_j$, $f$ and $g$ are implemented with neural network, Weibull$(k, l)$ is the Weibull distribution, which is reparameterizable (Zhang et al., 2018): Drawing $s \sim$ Weibull$(k, l)$ is equivalent to maping $s := l(-\log(1 - \epsilon))^{1/k}$, $\epsilon \sim$ Uniform$(0, 1)$. The SoftPlus applies $\log(1 + \exp(\cdot))$ nonlinearity to ensure the positive Weibull shape and scale parameters.

# C   DATASETS AND FURTHER EXPERIMENTS

## C.1   DATASETS

Our experiments are conducted on four widely-used benchmark text datasets including 20 News Group (20NG), Web Snippets (WS) (Phan et al., 2008), DBpedia (DP) (Lehmann et al., 2015), Reuters Corpus Volume 2 (RCV2) (Lewis et al., 2004) and one additional image dataset miniImageNet (Vinyals et al., 2016). For the text datasets, WS and DP are short documents, RCV2 and DP are large scale datasets. Since RCV2 owns multiple labels for one document, we follow the previous work and remove documents containing multiple labels in the second level, resulting in 0.15M documents. For the image dataset, miniImageNet is a subset of ImageNet (Russakovsky et al., 2015) dataset.

- **20NG**[1]: 20 Newsgroups consists of newsgroups post including 18,846 articles. We remove stop words and words with document frequency less than 100 times. We also ignore documents that contain only one word from the corpus. We follow WeTe of (Wang et al., 2022) and use the 6 super-categories as 20NG's ground truth and denote it as 20NG(6) in the clustering task.

- **WS**: Web Snippets is a short corpus that contains 12,237 web search snippets with 8 categories. There are 10,052 words in the vocabulary and the average length of a snippet is 15.

- **DP**[2]: DBpedia is extracted from Wikipedia pages. We follow the pre-processing process in Zhang et al. (2015), where the fields we used for this dataset contain title and abstract of each Wikipedia article.

- **RCV2**[3]: The original Reuters Corpus Volume 2 dataset consists of 804,414 documents. We here left documents that only contains single label at the second topic level, resulting in 0.15M documents totally, whose vocabulary size is 7282 and average length is 85.

- **miniImageNet** [4]: miniImageNet is a subset randomly sampled from ImageNet. In total, there are 100 classes with 600 samples of $84 \times 84$ color images per class. These 100 classes are divided into 64, 16 and 20 classes respectively for training, validation and testing.

A summary of text corpora statistics is shown in Table C. 1.

Table C. 1: Statistics of the datasets

|  | Number of docs | Vocabulary size(V) | average length | Number of labels |
|---|---|---|---|---|
| 20NG | 18,864 | 22,636 | 108 | 6 |
| DP | 449,665 | 9,835 | 22 | 14 |
| WS | 12,337 | 10,052 | 15 | 8 |
| RCV2 | 150,737 | 7,282 | 85 | 52 |

## C.2 BASELINES

We summary the baseline methods at Table. C. 2

| Model | VAE-based | HTM | transport-based | Word embedding |
|---|---|---|---|---|
| **LDA** (Griffiths & Steyvers, 2004) |  |  |  |  |
| **DVAE** (Burkhardt & Kramer, 2019) | ✔ |  |  |  |
| **ETM** (Dieng et al., 2020) | ✔ |  |  | ✔ |
| **NSTM** (Zhao et al., 2021) |  |  | ✔ | ✔ |
| **WeTe** (Wang et al., 2022) |  |  | ✔ | ✔ |
| **WHAI** (Zhang et al., 2018) | ✔ | ✔ |  |  |
| **SawETM** (Duan et al., 2021a) | ✔ | ✔ |  | ✔ |
| **BT-chain** |  | ✔ | ✔ | ✔ |

Table C. 2: Summaries of baseline models. HTM denotes the hierarchical topic model.

## C.3 MORE VISUALIZATIONS ON TEXT AND IMAGES

Similar to the main paper, we provide more visualizations of the learned topic hierarchies in Fig. C. 1 and Fig. C. 2.

---

[1]http://qwone.com/ jason/20Newsgroups

[2]https://en.wikipedia.org/wiki/Main_Page

[3]https://trec.nist.gov/data/reuters/reuters.html

[4]https://github.com/yaoyao-liu/mini-imagenet-tools

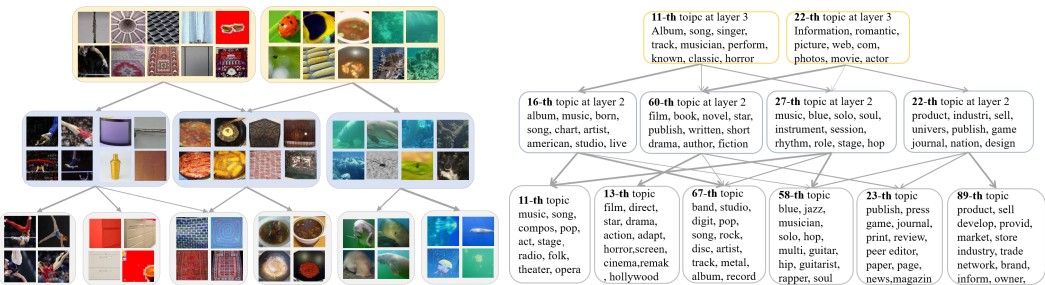

Figure C. 1: Topic hierarchies learned from miniImageNet (left) and DP (right).

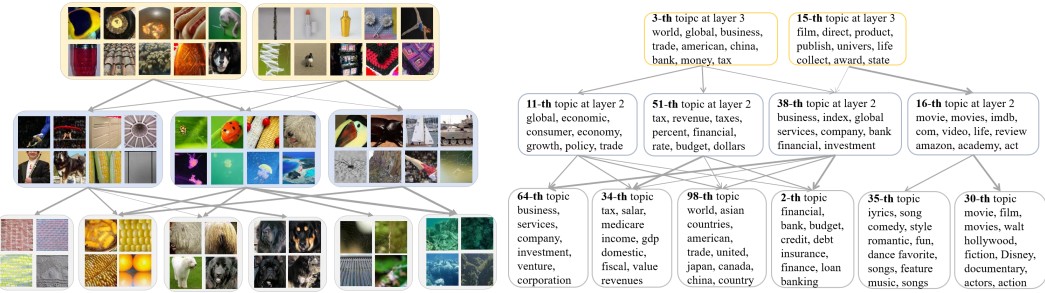

Figure C. 2: Topic hierarchies learned from miniImageNet (left) and WS (right).

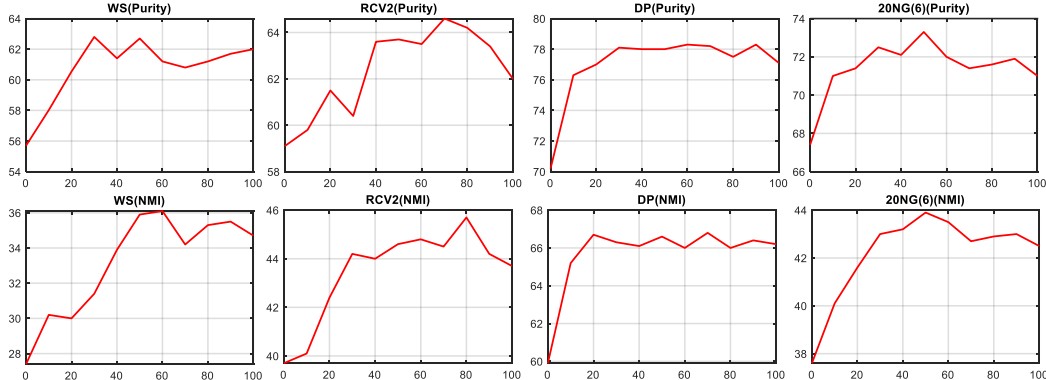

Figure C. 3: Clustering results with various $\beta$.

## C.4 HYPERPARAMETER SENSITIVITY

We fixed the hyperparameter $\beta = 50.0$ in all previous experiments for fair comparison. Here we report the document clustering results (Purity and NMI) with different hyperparameter settings on the four corpus in Fig. C. 3. Note that $\beta$ controls the weight of the cross entropy in Eq.6 in the mainscript. The regularized loss degenerates to the CT loss when $\beta = 0$. From Fig. C. 3 we find that the regularization term helps the document representations, as there is a significant improvement from $\beta = 0$ to $\beta = 10$; Besides, one can get better results than those reported in our experiments by fine-tuning $\beta$ for each dataset.

## D TIME COMPLEXITY ANALYSIS

The core computational module in BT-chain is the regularized CT loss between two adjacent layers $CT^{reg}(Q_j^{(l)}, Q_j^{(l+1)})$. It mainly contains the cost matrix, which has a time complexity of $\mathcal{O}(K_l K_{l+1})$ ($K_l$ is the number of topics at layer $l$), the bidirectional transport plan $\pi$, which has a time complexity of $\mathcal{O}(6K_l K_{l+1})$. We note that both of them have a linear complexity over the product of the number topics at two adjacent layers. For layer 0, we compute words within the target document rather than

all words in vocabulary, e.g. $K_0 = N_j$, $N_j$ is the number of words in document $j$, $N_j \ll V$. Thus the proposed BT-chain has an acceptable time complexity.

# E    METRICS

In our experiment, we report Topic Coherence (TC) and Topic Diversity (TD) to evaluate the learned topics from both the interpretability and diversity aspects. Given a reference corpus, TC measures the semantic relevance in the most significant words (top 10 words in our case) of a topic, which is computed by the Normalized Pointwise Mutual Information (NPMI) over the selected words of each topic Dieng et al. (2020):

$$f(w_i, w_j) = \left[ \log \frac{p(w_i, w_j)}{p(w_i)p(w_j)} \right] / \left[ -\mathrm{log} p(w_i, w_j) \right],$$

where $p(w_i, w_j)$ is the probability of words $w_i$ and $w_j$ co-occurring in a document and $p(w_i)$ is the marginal probability of word $w_i$, and both of them are estimated with empirical counts. Those models owing higher topic coherence are more interpretable topic models. TD measures how diverse the learned topics are. We define TD with the percentage of the unique word in the top 25 words of all topics Zhao et al. (2021). TD that closes to 0 indicates redundant topics; that closes to 1 means more diverse topics.

We also report Purity and Normalized Mutual Information (NMI) (Schütze et al., 2008) on clustering tasks to measure the performance of document representation. To compute Purity, each cluster is assigned to the class which is most frequent in the cluster, and then the accuracy of this assignment is measured by counting the number of correctly assigned documents and dividing by the total number of documents:

$$\mathrm{Purity}(S, C) = \frac{1}{N} \sum_k \max_j (s_k \bigcap c_j)$$

where $S = \{s_1, ..., s_K\}$ is the set of clusters, and $C = \{c_1, ..., c_J\}$ is the set of classes. $s_k$ and $c_j$ are sets of documents in cluster $k$ and class $j$, respectively. Higher purity means higher matching between $S$ and $C$. NMI is related to the information theory, which can be calculated as:

$$\mathrm{NMI}(S, C) = \frac{2I(S, C)}{H(S) + H(C)}$$

where $I(S, C)$ is the mutual information of $S$ and $C$, $H(S)$ is the entropy of $S$.

