# OpenReview forum: "BAT-Chain: Bayesian-Aware Transport Chain for Topic Hierarchies Discovery"
_ICLR.cc/2023/Conference — Submitted to ICLR 2023_

### Official Review · Reviewer_hEFb · 2022-10-25

**Confidence:** 4
**Correctness:** 3
**Technical Novelty And Significance:** 3
**Empirical Novelty And Significance:** Not applicable
**Recommendation:** 6

**Clarity, Quality, Novelty And Reproducibility:**

**Clarity**: the paper is mostly well written and easy to follow. There are some minor issues that would be benefited by polishing of the text. Details below.

**Quality**: The method and experiments appear to be sound.

**Novelty**: Considering the question above is resolved positively, the paper appears to be novel.

**Reproducibility**: The paper discusses the details of implementation and provides the source code, therefore, the results should be reproducible. However, some details are missing, see e.g. 12 and 13 points below.

Specific comments/suggestions:
1. Introduction. Second paragraph. “Despite the success…, the learning … is done by Bayesian posterior inference…” – odd wording if to ignore the “which” statement that follows.
2. Introduction. Second paragraph. “Another concern comes from the pure likelihood maximization” – not sure what “pure likelihood maximization” is supposed to mean here.
3. Introduction. Second paragraph. “even acute” – “even more acute”? Otherwise, it sounds a bit odd.
4. Introduction. Third paragraph. “both distributions” -> “the two distributions”
5. Introduction. 4th paragraph. “This results in a more flexible and efficient method” – more than what?
6. Section 2. Second paragraph. “To measure such two discrete distributions” – to measure a distance between such two discrete distributions?
7. End of page 3. “HTMs” – acronym is not defined
8. Section 3.2. First paragraph. “With the fact that …” – unfinished sentence
9. Experiments. “For the ETM, NSTM and WeTe that work with word embeddings, we use the pre-trained GloVe vectors” – the proposed work also work with word embeddings, does it use the same GloVe vectors?
10. Experiments. Why [100, 64, 32] those numbers were chosen for topic numbers?
11. Experiments. Metrics Purity and NMI are unclear from the description.
12. Figure 2. What is x-axis?
13. It is not clear how could links between topics on different layers be computed? For example, those links in Figure 3
14. First line on page 13. “where we specific transport cost”: “we specify”?

Minor
1. Redundant commas:
    * Introduction. 4th paragraph. “hierarchical, topical, distributional view”
    * Introduction. 5th paragraph. “is a straightforward, and effective approach”
2. Section 2. First paragraph. “states with the satisfy” – redundant “the”
3. Section 3.2. First paragraph. “corresponds to the learning” – redundant “the”
4. Section 3. Last paragraph. “with THE Bayesian flavor”
5. Footnote 1. “are not be applied” -> “are not applicable”


**Strength And Weaknesses:**

**Strengths**:
* Tackling topic hierarchy discovery
* Rather extensive experiments
* Easy to follow text

**Weaknesess**:
* (minor) Polishing is required (some examples below) but should be doable during the rebuttal stage
* (minor) Although there is nothing apparently wrong with the paper, it doesn’t excite too much. Topic models is a well-researched area. The paper takes the previously proposed settings with word embeddings and topics living in the same space as embeddings with inference based on conditional transport. The paper then extends these ideas to discover topic hierarchy. I can’t say there is not enough novelty in this, but it is just not exciting as looks like a kind of natural extension of those previous settings – it is probably easy to say when it has been done already in this paper, but still. The paper does make an attempt on bringing something else by discussing the link with Bayesian learning – which is a great way to “spice” things up. However, and this is a separate weakness on its own, this link to Bayesian learning discussion is a little bit weak. It would be great to see it more elaborated, with probably some illustrative examples. The whole link is currently based on the fact that in Bayesian learning we have likelihood from the data (i.e., words) and prior from topics of the higher levels and in the proposed model training of a topic on each level is done by transferring the information from below (which is words at the end) and above (which is topics of the higher levels).
* The link to Bayesian learning is not convincing or elaborated enough (see details above). I believe the paper should either elaborate on this or downplay this link in the text (it is even used in the title which is not justified in the text).

**Question (which may lead to weaknesses)**:

The provided code looks very similar to the code for the WeTe model, moreover, it is even called WeTe. First, WeTe initial code should be acknowledged. Second, could the authors please briefly discuss in their rebuttal what are the main differences in the code?


**Summary Of The Paper:**

The paper proposes a topic model that discovers a topic hierarchy and works with word embeddings. Inference in the model is done by using conditional transport. The inference is also linked to the Bayesian inference.

**Summary Of The Review:**

I believe the question about code similarity would be resolved (there is nothing wrong in using the existing code as a base), then the only real issues left would be overselling of the Bayesian link, which is fixable by downplaying it in the text or elaborating on it to be more solid, and this lack of excitement about the method, but this is not really an issue and may be just me. If to look at the facts, this is a solid piece of work, extending the existing topic modeling approach for discovering topic hierarchies with rather extensive empirical study to support this extension.

---

> ### Author Response · Authors · 2022-11-18
> **Thanks to reviewer hEFb for the questions and feedback**
>
> Thank you for your polishing suggestions, detailed comments, and the positive feedback, which help us to improve the quality of our paper. Below we answer your main concerns:
>
> **Q1**
>
> We follow your suggestions and downplay the Bayesian link in the revision. We have changed the title to ''BT-chain: Bidirectional Transport Chain for Topic Hierarchical Discovery'' and modified related statements. Please move to the general response for more detailed discussion.
>
> **Q2**
>
> Thank you for the kind reminder. We will acknowledge WeTe in the revision.
>
> We implement our BT-chain based on the released code of WeTe. The code of WeTe provides several useful packages to compute the CT distance between two adjacent layers, which are basic components of our BT-chain. The main difference between BT-chain and WeTe comes from the training algorithm implemented in Trainer.py. Specifically, BT-chain focuses on solving the hierarchical topic modeling problem and requires an upward-downward layer-wise training algorithm to optimize its parameters that WeTe doesn’t contain, although the two models shared the same basic computing function and other helpful functions (e.g., the python function of data loader, metrics calculation).
>
>
> **Q3**
>
> For the models that work with word embeddings, including ETM, NSTM, WeTe, and our BT-chain, we load the pre-trained GloVe vector for a fair comparison. We have highlighted this in the revision.
>
> **Q4**
>
> We view the topic numbers at each layer as the hyperparameters in BT-chain and set them as K= [100,64,32] for all hierarchical topic models empirically. The choice of K often satisfies k_1 > k_2 >,,,>k_l >,,,>k_L in hierarchical setting.
>
> **Q5**
>
> The Purity and normalized mutual information (NMI) are two widely-used metrics in clustering task. To compute purity, each cluster is assigned to the class which is most frequent in the cluster, and then the accuracy of this assignment is measured by counting the number of correctly assigned documents and dividing by the total number of documents:
>
> $ purity(S, C) = \frac1N \sum_k \underset {j}{max} |s_k \bigcap c_j| $
>
> where $S={s_1, ...,s_K}$ is the set of clusters, and $C={c_1,...,c_J}$ is the set of classes. $s_k$ and $c_j$ are sets of documents in cluster k and class j, respectively. Higher purity means higher matching between S and C.
>
> NMI is related to the information theory:
>
> $NMI(S,C) = \frac {I(S,C)}{[H(S) + H(C)] / 2}$
>
> where I() is the mutual information, and H() is the entropy.
>
> **Q6**
>
> We compare BT-chain with other hierarchical topic models under different topic layers L at Fig.2, and x-axis denotes the topic layer L. We have highlighted this in the revision.
>
> **Q7**
>
> The links between topics denote the connection weights of the topic at two adjacent layers, which is calculated according to the cosine similarity of corresponding topic embeddings (e.g., $\phi$ in Eq.4)

---

> > ### Comment · Reviewer_hEFb · 2022-11-18
> > **Thank you for your response**
> >
> > Thank you for your response.
> >
> > You have clarified my potential concerns about the code. Thank you and thank you for answering other questions.
> >
> > However, could you please still address two of the questions? They are not major, but it would be better to reflect their answers in the text.
> >
> > * Why [100, 64, 32] those numbers were chosen for topic numbers?
> >
> > When you say "K= [100,64,32] for all hierarchical topic models empirically" what exactly do you mean? And I believe other readers would also be interested to know that so it should appear in the text.
> >
> > * Metrics Purity and NMI are unclear from the description.
> >
> > Thank you for clarification on the metrics, but I meant that they should appear in the text. It can be appendix, but it is better to have them. Moreover, you do not even have references for them, and sometimes people use different implementations of metrics with the same name.

---

> > > ### Author Response · Authors · 2022-11-18
> > > **Re: Thank you for your response**
> > >
> > > Thank you for your kind reply.
> > >
> > > - The topic numbers at each layer in hierarchical topic models are often treated as the hyperparameters. To make the fair comparison, we set K=[100,64,32] for all three-layer baseline models in our experiment. We follow your advice and will highlight this in our revision.
> > >
> > > - We follow your advice and will add the missing references and metrics section in the appendix.
> > >
> > > We sincerely hope our responses can answer your questions and improve your evaluation.
> > >
> > > We're happy to answer your further questions.

---

### Official Review · Reviewer_8ZrW · 2022-10-26

**Confidence:** 4
**Correctness:** 3
**Technical Novelty And Significance:** 3
**Empirical Novelty And Significance:** 3
**Recommendation:** 5

**Clarity, Quality, Novelty And Reproducibility:**

Clarity: the paper is generally reasonably easy to follow.

Quality: while the modeling formulation is novel and interesting, the proposed algorithm is not well justified and may not succeed at optimizing the model's overall objective function.

Novelty: this work combines ideas from two different sub-areas of topic modeling work. As such, the contribution is sensible if somewhat incremental.

Reproducibility: code was provided in the supplementary, which would help with reproducibility.

**Strength And Weaknesses:**

Strengths:

- The proposed formulation is sensible and elegant, combining ideas from two previously disparate research directions in a clever way to develop a sophisticated model that leverages the advantages of both.

- The conditional transport approach is expected to provide computational advantages over the previous Bayesian probabilistic modeling approaches.

- The application of the method to image data as well as text was quite interesting.

- It's nice that the method can leverage TFIDF data as input instead of bag of words data. I have not seen topic models that do this. It probably improved performance. (It would however be worth doing a comparison to bag of words to show this, and this was not done in the present manuscript.)

Weaknesses:

- While the modeling formulation is quite compelling, the proposed training algorithm is not well justified. Two update steps are proposed: an "upward warming up" used in an initial phase and a "downward refining" step (more accurate?). The upward phase ignores half of the terms in the loss function, so is very approximate. The downward phase updates one layer at a time, leveraging the approximate solutions from the layer below computed in the first pass. The layers are not revisited, so every update depends on the approximate first-pass solution from the layer below. This may be a practical approach, but it is clearly not optimal.

Potentially, a better approach might be to perform stochastic gradient updates where the stochasticity is over the loss terms from all of the layers together. I.e., in each stochastic gradient iteration, choose a random document j and a random layer l and take a stochastic gradient step updating Equation 3 with the chosen j and l. This algorithm is a stochastic gradient optimizer for the overall loss function, which is easier to justify than the proposed algorithm.

- The "Bayesian flavor" interpretation of the method is a bit of a stretch. I see no substantive connection to Bayesian updating, as the paper claims.

- Perplexity results are not given as it is stated the metric is "unsuitable for CT based topic models." Why is it unsuitable? Is it because CT-based topic models look bad under this metric? (This isn't a deal-breaker since the TC and TD results were good, and these metrics are arguably more important. But there needs to be a better explanation of it.)

- Minor point: A "BAT-chain-U" baseline is mentioned but I do not see results for it in any of the tables.

**Summary Of The Paper:**

This paper proposes a hierarchical topic modeling approach based on conditional transport, combining previous research on hierarchical topic models with recent research on conditional transport. The idea is that documents are represented by discrete distributions over word embeddings, topics at each level of the hierarchy are also represented via embeddings, and similarity between topics at different levels of the hierarchy is enforced via a conditional transport cost between each document's distribution-over-embedding representations at adjacent layers/levels of the hierarchy.

**Summary Of The Review:**

The proposed CT-based hierarchical topic model is an interesting combination of previous and recent ideas into a new sophisticated model with nice and potentially useful properties. While the work tackles a worthy problem and proposes an elegant formulation, the training algorithm is heuristic in nature and its ability to optimize the stated objective function is not theoretically motivated. It is likely that a modification of the proposed method could be on a firmer theoretical footing (see the review above). A couple of other claims in the paper are unsupported as well (E.g. the "Bayesian" interpretation, perplexity metric is not appropriate). Overall, the research is promising but it needs a bit more work before it is ready for publication.

---

> ### Author Response · Authors · 2022-11-18
> **Thanks to reviewer 8ZrW for the questions and feedback**
>
> Thank you for the valuable suggestions and appreciating the elegant formulation we proposed in this paper. Below we answer your concerns:
>
> **Q1**
>
> In fact, our baseline, WeTe, uses the normalized BoW vector as the input. In our submitted manuscript, we have compared our single-layer BT-Chain with WeTe in Table 1, where the single-layer BT-Chain degrades to WeTe but takes the tfidf data as input. The proposed BT-Chain achieves a slight improvement, which demonstrates the efficiency of introducing word weights via tfidf.
>
> **Q2**
>
> Thank you for your detailed training algorithm suggestion. The core idea behind BT-chain is to minimize the total transport cost among the L discrete distributions of each document in the training corpus. The training algorithm plays a core role in BT-chain. Unfortunately, it is intractable due to the three need-to-be-learned discrete distributions in Eq.3. The proposed upward-downward training algorithm first transports the input information from layer 0 to layer L by minimizing the forward CT distance via the upward path. This makes sure $Q^{(l)}$ at each layer $l$ has relative accuracy estimation to guide the downward refining. The training algorithm you suggested is similar to our downward refining step. The main difference is that Eq.3 is optimized with a layer-wise strategy in the downward refining step, while a layer-sampling strategy in yours. Though the upward path is a practical alternative, our proposed method is more efficient and achieves better results empirically.
>
> **Q3**
>
> We follow your advice and have changed the title to ''BT-chain: Bidirectional Transport Chain for Topic Hierarchical Discovery'' and related statements in the revision. Please move to the general response for detailed discussions.
>
> **Q4**
>
> Perplexity is designed to evaluate the prediction performance of a model. Unfortunately, those transport-based NTMs mainly focus on mining high-quality topics, where topics are learned by minimizing the transport cost instead of maximizing the log-likelihood, which is important to achieve better perplexity results. Therefore, it is unfair and unsuitable for CT-based models to compare the perplexity results with such Bayesian generative topic models. Moreover, perplexity is not an appropriate measure of the topic quality and sometimes can even be contrary to human judgments, as previous studies found [1][2]. We further explained the reason in our revision.
>
> We follow your advice and report the perplexity (PPL) of ETM, SawETM, and BT-chain on 20NG below (the number in brackets denotes the topic layer L):
>
> |  Method  | ETM |  SawETM (1) |  SawETM (2)  |  SawETM (3)  | BT-chain (1)  | BT-chain (2) | BT-chain (3) |
> |:--------:|:------:|:----:|:----:|:----:|:----:|:----:|:----:|
> | PPL |  3082.8  | 2679.4 | 2357.7 | 2195.3 | 3214.4  | 3084.6 | 2967.1 |
>
> It is not surprising that ETM and SawETM achieve better perplexity than BT-chain due to their log-likelihood objective function. We find that the ppl score of BT-chain gets better as the number of layer L increases, which demonstrates the motivation of BT-chain. Besides, BT-chain (3) performs a little better than ETM. This denotes that BT-chain is not only able to discover higher-quality topics and better document representations, but also achieve comparable prediction performance with Bayesian generative topic models.
>
> [1] J. Chang, S. Gerrish, C. Wang, J. L. Boyd-Graber, and D. M. Blei. Reading tea leaves: How humans interpret topic models. In NeurIPS2009.
>
> [2] L. Yao, Y. Zhang, B. Wei, Z. Jin, R. Zhang, Y. Zhang, and Q. Chen. Incorporating knowledge graph embeddings into topic modeling. In AAAI2017.
>
> **Q5**
>
> We propose BT-chain-U, a variant of BT-chain, that only uses the upward warming-up path. We report the results of BT-chain-U at Fig.2 in the manuscript.

---

### Official Review · Reviewer_G6X6 · 2022-11-03

**Confidence:** 4
**Correctness:** 3
**Technical Novelty And Significance:** 3
**Empirical Novelty And Significance:** 3
**Recommendation:** 5

**Clarity, Quality, Novelty And Reproducibility:**

(1) Clarity: The paper is cogent and polished. In detail, the delineation in the methodology section is sound and intuitive. I can grasp the mechanism of the bidirectional training strategy and the conditional transport objective.

(2) Originality/Novelty: The paper presents a novel framework for hierarchical topic modeling based upon the optimal transport theory basis.

(3) Quality/Significance: The experimental section indicates the improvement over recent SOTA baselines on diverse datasets. This demonstrates the efficiency and robustness of the proposed method.

(4) Reproducibility: The submission provides source code for re-implementation, which would help the reproducibility.

**Strength And Weaknesses:**

**Strengths**:

The authors introduce a novel solution for discovering topic hierarchy. The paper is generally smooth to follow, and the experiments can prove the prowess of conditional transport for hierarchical topic modeling.

**Weaknesses**:

From my perspective, there exist several drawbacks in the Introduction, Methodology, and Experiment sections:

(1) In the introduction section, I cannot see clearly why reparameterization and KL divergence hurt AVI, especially if AVI and CT are adapted to hierarchical topic modeling, and what merits does CT provide to address the advantages of AVI?

(2) The motivation of CT for hierarchical topic modeling is quite obscure. The introduction only explained that CT was applied because previous single-layer topic modeling had not continued to explore its usage for hierarchical circumstances.

(3) The methodology section mentioned that ``A TOPIC in layer 1 is expected to capture more general information than A WORD in layer 0``. Comparing a topic with a word sounds unnatural to me. Such interpretation somehow makes the hypothetical basis for CT topic hierarchy less persuasive.

(4) The desiderata of the proposed framework is that the upper layers will capture more general topics than the lower layers. Nevertheless, Figure 3 does not clearly indicate the aforementioned relationship. In contrast, the illustrated upper topics have more tendency to display semantic proximity to the lower ones, instead of generalization relation. Additionally, the topics appear to be incoherent as well. For example, why do topic 52 and 24 comprise ``philosopher`` and ``medical``, respectively?

**Summary Of The Paper:**

This paper proposes BAT-Chain, a hierarchical topic model leveraging multi-layer conditional transport (CT) theory to seize topic structures. The proposal is inspired by previous application of CT to the single-layer topic modeling.  The authors conduct extensive experiments in both textual and visual settings to showcase the merits of their method.

**Summary Of The Review:**

A novel approach is proposed to tackle the multi-layer topic discovery problem. The elucidation of the CT framework is rational and comprehensible. However, there is some degree of vagueness in the motivation and hypothetical discussion. Moreover, the qualitative analysis is also not convincing, which drives me to my overall judgement.

---

> ### Author Response · Authors · 2022-11-18
> **(1/2) Thanks to reviewer G6X6 for the questions and feedback**
>
> Thank you for the detailed review and appreciating the novelty of our solution on hierarchical topic discovery. Below we address specific questions.
>
> **Q1**
>
> We consider that the reparameterization trick and KL divergence can be tricky in neural topic models (NTMs). As we all know, AVI usually has a classical assumption about hidden variables that obey a Gaussian distribution. Benefiting from the Gaussian distribution, AVI designed in this way has the desired reparameterization trick and analytical KL divergence, the critical factors for the success of AVI. However, for topic models, Dirichlet and Gamma distributions are usually used, which are not directly reparameterizable. Therefore, complex sampling schemes or approximations are needed to use AVI for NTMs. These approximations may lead to non-analytical KL divergences, which further complicates the learning algorithm. Please see Section 3.1 in [1] for more detailed discussions. In hierarchical topic models, the learning algorithm of AVI can be much more complex. Different from the framework of AVI, ours is based on CT, which is defined by the bidirectional (forward and backward) transport cost between the source and target distributions. Therefore, our proposed method can bypass the tricky parts of AVI and learns hierarchical topic models in a more straightforward and flexible way, i.e.,  an efficient upward-downward optimizing strategy under the recent conditional transport theory.
>
> [1] Zhao, H., Phung, D., Huynh, V., Jin, Y., Du, L., \& Buntine, W. Topic Modelling Meets Deep Neural Networks: A Survey.
>
>
> **Q2**
>
> Our paper is mainly motivated by the development in conventional Bayesian topic modeling from single-layer models like LDA to hierarchical models like Nested hierarchical Dirichlet processes. Being hierarchical can provide more interpretability and a deeper understanding of the semantics in documents, which is the main goal of topic models. Inspired by this observation, various hierarchical topic models are gradually developed, most of which however are based on AVI frameworks. And we have discussed the limitations of applying AVI into NTMs. Therefore, we in this paper propose a new method based on CT frameworks due to its advantages in mining single-layer topics. However, extending a single-layer CT method to a multi-layer one is non-trivial, which requires new formulations and learning algorithms. That motivates our paper. The proposed BT-Chain aims to bridge the gap and provides an efficient attempt for the hierarchical topic modeling community. Although both are based on CT theory, our BT-Chain is distinct from WeTe in terms of technical details and tasks. BT-Chain constructs a series of discrete distributions over the corresponding embedding spaces for each document, exploring the hierarchical document representations. In addition, an efficient and end-to-end training algorithm is then developed carefully to train BT-Chain in terms of CT distance.
>
> **Q3**
>
> Thank you for your valuable suggestions, we have revised related descriptions in the rebuttal revision.

---

> > ### Author Response · Authors · 2022-11-18
> > **(2/2) Thanks to reviewer G6X6 for the questions and feedback**
> >
> > **Q4**
> >
> > As the topic numbers decrease from layer 1 to layer L, the upper layers in the BT chain typically capture more general topics which gradually summarize the information from the lower layer with few topics. As a result, each topic in the upper layer tends to mix semantically close topics at the lower layers. We visualize the top-n words of each topic according to their cosine similarity scores in Fig.3. The ''more general topic'' refers to the mixtures of refining topics and does not always contain more general words.
> >
> > As for the incoherent topics (e.g., topic52 and topic24), we note that our proposed model learns topic hierarchies according to the given corpora. ``It is not surprising that topic52 contains ''philosopher'' if several articles contain philosopher and some other words close to ''medical'' ``.
> >
> > **There are 17 documents in 20NG that mention the words ''medical'' and ''philosophical''.**
> >
> > For example, in 15516-th documents: ''..., anyone else asking for ``medical`` information on some subject could ask specific questions, as no one is likely to type in a textbook chapter covering all aspects of the subject.  If you are looking for a comprehensive review, ask your local ``hospital`` librarian. ... but whoever wishes to become a ``philosopher`` must learn not to be frightened by absurdities... ''
> >
> > **There are 143 documents in 20NG that mention the words ''books'', ''research'', ''systems'', and ''medical''.**
> >
> > For example, in 5226-th document: '' ..., Since this is a major portion of ``medical`` education, ..., Has anyone had luck with any particular ``books``, memory ``systems``, or : cheap ``software``?     : Can you suggest any helpful organizational techniques?''
> >
> > In 9302-th document: ''..., ``Biomedical research`` doesn't make any basic assumptions that aren't the same as any other discipline of scientific ``research``.  That is, that you make empirical observations, form an hypothesis and test it.  Modern ``medicine`` has much more to do with biochemistry ... ''.

---

### Author Response · Authors · 2022-11-18
**General Response and summary of Updates to Manuscript**

We thank all reviewers for their time and valuable feedback to improve the work. We have uploaded a rebuttal version with the major changes marked in blue based on your feedback. The typos, redundant statements, and the Bayesian-aware interpretation have been addressed in the revised version. We believe that this has made our paper stronger.  We are hopeful that the new changes resolve your concerns, thus improving your opinion of our work. Below, we first address the common concerns of all reviewers and then answer specific questions.

One of the common concerns among reviewers is the overemphasis on Bayesian interpretation.

One of the key ideas of Bayesian topic models is the use of prior distributions of document-topic proportions and topic-word proportions. Prior distributions provide ''prior knowledge'' to a model, e.g., Dirichlet distributions provide sparsity in document-topic proportions and topic-word proportions (a key success of LDA over PLSA). More informative prior distributions can provide richer prior knowledge, e.g., prior distributions informed by word embeddings for topic-word proportions such as in ETM. The proposal of hierarchical topic models also follow this general idea as higher-layers are prior distributions of lower-layers. From this point of view, we believe that our method is inspired by and shares the same thinking with Bayesian models. That is why we named our method ''Bayesian-aware'' previously.

But we agree that technically, our method less aligns with Bayesian models and it might not be appropriate to highlight the Bayesian flavor of our method in the paper. According to your suggestion, we have revised our paper, including changing the title to '' BT-chain: Bidirectional Transport  Chain for Topic Hierarchical Discovery'' and other related statements.

---

### Decision · Program_Chairs · 2023-01-20

**Decision:**

Reject

**Justification For Why Not Higher Score:**

The technical contribution of the paper is somewhat limited.

**Justification For Why Not Lower Score:**

N/A

**Metareview: Summary, Strengths And Weaknesses:**

This paper proposes BAT-Chain, a novel model to mine hierarchical topics and document representation under the conditional Bayesian-aware transport framework. Extensive experiments on text corpora show the effectiveness of the proposed method.

In general, the paper is well-written and easy to follow. The experimental results demonstrate the efficiency and robustness of the proposed method. The reviewers raised a few concerns about the technical details, clarification, and novelty of the paper. The authors addressed some of them but the reviewers are not fully convinced, especially about the technical contributions of the paper.

**Summary Of Ac-Reviewer Meeting:**

The reviewers raised a few concerns about the paper's writing, technical details, clarification, and novelty of the paper. Even though the paper can be polished in writing and technical details, the reviewers agreed that the novelty and technical contribution of the paper is somewhat limited, so they voted for the rejection of the paper. I agree with that suggestion.